# Meta-Analysis of Gene Popularity: Less Than Half of Gene Citations Stem from Gene Regulatory Networks

**DOI:** 10.3390/genes12020319

**Published:** 2021-02-23

**Authors:** Ionut Sebastian Mihai, Debojyoti Das, Gabija Maršalkaite, Johan Henriksson

**Affiliations:** 1Molecular Infection Medicine Sweden (MIMS), Umeå Centre for Microbial Research, Department of Molecular Biology, Umeå University, 901 87 Umeå, Sweden; ionut.sebastian.mihai@umu.se (I.S.M.); debojyoti.das@umu.se (D.D.); marschalka@gmail.com (G.M.); 2Industrial Doctoral School, Umeå University, 901 87 Umeå, Sweden; 3National Clinical Research School in Chronic Inflammatory Diseases (NCRSCID), Karolinska Institutet, 171 77 Solna, Sweden

**Keywords:** gene, Matthew effect, biological feature, genomics, machine learning, linear model, gene regulatory networks

## Abstract

The reasons for selecting a gene for further study might vary from historical momentum to funding availability, thus leading to unequal attention distribution among all genes. However, certain biological features tend to be overlooked in evaluating a gene’s popularity. Here we present a meta-analysis of the reasons why different genes have been studied and to what extent, with a focus on the gene-specific biological features. From unbiased datasets we can define biological properties of genes that reasonably may affect their perceived importance. We make use of both linear and nonlinear computational approaches for estimating gene popularity to then compare their relative importance. We find that roughly 25% of the studies are the result of a historical positive feedback, which we may think of as social reinforcement. Of the remaining features, gene family membership is the most indicative followed by disease relevance and finally regulatory pathway association. Disease relevance has been an important driver until the 1990s, after which the focus shifted to exploring every single gene. We also present a resource that allows one to study the impact of reinforcement, which may guide our research toward genes that have not yet received proportional attention.

## 1. Introduction

One of the current great challenges in biology is integrating knowledge into a coherent model, thus allowing predictions to be made. However, this quest heavily relies on our understanding of all the different features that define our biological question. How well do we understand the different features, and has the manner or motivation for study affected our conclusions about them? As systems biologists, we wondered if we could somehow address these questions based on the intrinsic properties of the genes. Similar studies have previously addressed this question, making a great contribution in highlighting social features (funding, transitioning to principal investigator status, model organism and scientific literature database availability) and a plethora of physicochemical properties of protein-coding genes [1]. However, the amount of literature about factors behind gene popularity integrating biological feature information yielded by NGS-derived datasets, CRISPR-screens, gene regulatory (GRNs), and protein–protein interaction (PPis) networks is limited.

Scientometry is the discipline that studies scientific and technologic literature from the quantitative perspective [2]. From scientometry it is known that literature is usually skewed to cover some subjects at a much greater depth than others. Various statistical distributions seem able to explain current and past publication trends, proposed to follow laws such as those of Bradford and Lotka, or the Pareto distribution [3]. It is, however, not clear what constitutes a “subject” nor how generally consistent these principles are. Pareto-like distributions are generated in systems having the Matthew effect, in other words, a positive feedback loop where “the rich become richer”. In scientific research, a few critical discoveries nucleate fields of which some grow much faster than others. Throughout this paper we will call this effect “reinforcement”.

In this study we consider individual genes as “subjects” and show that literature follows a Matthew-like principle. We theorize that this is because it is easier to study genetics once a few “reference genes” have been discovered and studied. However, this also means that the number of papers might mainly reflect a social process of discovery rather than reflect the real relevance of the genes to the subject of interest.

To correlate citations with social driving force versus a gene’s biological relevance, we made use of unbiased datasets that may suggest a gene to be perceived as important, including single-cell RNA-seq data, protein–protein interactions, as well as CRISPR screens (Figure 1a). The model is by necessity semi-qualitative—there are multiple ways to encode the features mathematically. Furthermore, it will always be possible to add further features that could be of relevance. Thus, the results need to be interpreted considering the model formulation. We try to avoid demarcating different sources as “social” or “biological”, but our choice of biological factors unavoidably reflects our own view of “importance”. Our unit of study is one gene, but we could have considered transcript isoforms, post-translationally modified proteins, or protein domains. These are all valid alternative objects of enquiry but outside our scope. We have largely ignored how different experimental methodologies have impacted citations (e.g., proteomics vs. transcriptomics). Finally, gender, class, and ethnicity could all be included as social reinforcement factors, but here we were mainly interested in the overall balance of social vs. biological factors.

If we consider the initial reporting date (or time) since the discovery of a gene as the main social component predictor, then roughly 25% of the papers are a result of social reinforcement. Gene expression level is the second strongest indicator, followed by markers of disease relevance to a lesser extent. We believe that further use of unbiased data generation methods will widen the set of genes considered and hopefully enrich our understanding of cell biology.

## 2. Materials and Methods

### 2.1. Pubmed Data Retrieval and Pre-Processing

Genes and PMIDs were retrieved using the publicly available FTP service (Available online: ftp://ftp.ncbi.nih.gov/gene/DATA/gene2pubmed.gz (accessed on 29 April 2020)); released on 16 December 2019). Only mouse and human IDs were retained. Mouse ENSMUSG gene IDs were processed and converted to human Ensembl IDs (ENSID) using BioMart. Further Pubmed article metadata was downloaded (ftp://ftp.ncbi.nlm.nih.gov/pubmed/baseline (accessed on 29 April 2020)). A custom Java program (available on our Github) was used to extract date of publication and PMID.

Cell type keywords were defined semi-manually based on the cell type annotation in the Tabula Muris dataset. MeSH terms could have been a better choice, but literature also suggests they are not commonly used [4]. Poorly represented cell types were removed or merged with other categories. The mapping cell type—{PMIDs} was created by searching Pubmed for keywords (see Supplementary File S1). This association was used along with the gene-PMIDs mapping to create cell type-specific lists of papers.

The measure #citation was defined as log10 (1 + number of papers for a gene), either total paper counts or subsetted for one cell type. We also tried to use the rank (number of papers per gene) as a measure, hoping it would better even out the statistical distribution; however, because too many genes had similar (low) number of citations, many ranks were tied, and we decided against the use of this measure.

### 2.2. Citation Distribution Analysis

We investigated distributions for several genes, and they followed similar trends. Comparison with the exponential distribution was made with fitdistr() from the R MASS package [5]. Comparison with the Pareto distribution was made with the R ParetoPosStable package.

### 2.3. Gene Family Analysis

Gene symbol nomenclature for the mouse genome (*Mus musculus*) was extracted from Tabula Muris datasets. Only gene symbols with a structure containing any combination of characters from a to z (English alphabet) followed by any combination of digits from zero to nine were analyzed (structures similar to Abc followed by 1, 2, 3, or specified as a regular expression: (a-zA-Z) + (0–9) +), with a total of 18,330 unique gene symbols. The digits make up f_FI_, while #citation of the gene with f_FI_ = 1 was used as f_founder_. For founders themselves, f_founder_ was set to N/A. f_FI_ was capped at 30, and this value was also used for genes that do not have a founder, following our nomenclature. In the total model, f_founder_ was set to the median value whenever N/A.

### 2.4. Gene Homology Analysis

All the mouse protein sequences were obtained from Uniprot (ID UP000000589). A Java script was used to reduce the FASTA header name to just the gene symbol, and only genes included in the Tabula Muris count tables were considered. The command “blastp -db uniprot.fa -query uniprot.fa -out results_prot.out -outfmt 6” was used to do all-against-all mapping (version 2.6.0+) [6]. Only the highest blastp “pident” score for each pair of genes was retained.

Two methods were used to enrich the graph for edges between the most similar genes. First, only the top 3 edges were retained (largely to speed up following the calculations). Second, a triangle-inequality method was used, inspired by ARACNe DPI [7]. For genes X, Y, and Z, we compared their protein identity I. If I_xz_ > I_xy_ and I_yz_ > I_xy_, the edge X–Y was removed. Informally, this means that Z was sufficiently well-matched to intermediate X–Z–Y that the link X–Y could be considered superfluous. This algorithm was implemented in Java. The #citation of the connected genes then defined f_homology_.

### 2.5. Gene Expression Analysis

Gene count tables from the Tabula Muris [8] were retrieved from https://tabula-muris.ds.czbiohub.org/ (accessed on 29 April 2020). We used the “FACS sorted, SMART-Seq2 RNAseq libraries” as the depth appeared better for co-expression analysis, so we used these libraries throughout for consistency. Based on the existing cell type annotation, the number of cells in each tissue was counted. The tissue with the largest number of cells matching a given cell type was designated “the primary tissue”. The average counts were calculated for each cell type in their primary tissue (by focusing on one tissue we needed not consider batch effects). f_exp_ was defined as rank (expression level).

### 2.6. Gene Co-Expression Network Analysis

The Tabula Muris RNA-seq count table for different cell types was used again. Instead of the average, single cell counts from the primary tissue were retained. The counts were rescaled as log10 (1 + count). The first 6 PCA components were calculated by prcomp_irlba. A projection was made with UMAP [9]. The k-nearest neighbor (KNN) graph was calculated with k = 10. f_coexp_ was defined as average(#citations) of these neighbors.

### 2.7. Protein–Protein Interaction (PPI) Network Analysis

We downloaded HuRI.tsv (available online: http://www.interactome-atlas.org/ (accessed on 29 April 2020) [10]. As this data were already in the form of a network, we could use them directly without intermediate processing. The triangle inequality was not applied, as we assumed the dataset to only consider direct interactions. f_PPI_ was defined as average(#citations) of the neighbors of each gene.

### 2.8. Gene Essentiality Analysis

We downloaded Appendix A from the CRISPR screen study online supplement [11] and used the column “% Dependent Cell lines”. The global essentiality score as defined by their fuzzy set AdAM algorithm was within the range (0, 100), and thus we used it directly as f_essential_. For genes not included in the dataset, we set the corresponding value to the median. We also attempted to generate cell type-specific essentialities by manually curating the cancer cell line types and comparing them to Tabula Muris tissues; however, we usually did not manage to clearly decide which exact cell type was the origin and so this was not used in the end.

### 2.9. Gene Chromatin Proximity Analysis

The mouse genome GRCm38.97 GTF-file was downloaded from Ensembl. Features of the type “gene” were retained, and gene positions were calculated as (from-to)/2. The gene symbol was extracted from the attributes field. The coordinate table was merged with the cell type-specific paper counts. For each chromosome and cell type, the features were sorted. Then for each gene, the closest other genes obtained and f_chromatin_ were defined as the average #citations of these.

### 2.10. GWAS Analysis

The file gwas_catalog_v1.0-associations_e98_r2020-03-08.tsv was downloaded from the EBI GWAS catalog. We considered as targets those genes in the column “REPORTED.GENE.S.”. Intergenic SNPs were removed. The smallest *p*-value for any SNP was calculated but capped at 10^−40^. f_GWAS_ was defined as –rank (*p*-value) such that high positive values implied high relevance.

### 2.11. COSMIC Analysis

We downloaded the file CosmicGenomeScreensMutantExport.tsv.gz and used a Java program to extract the number of mutations and length (amino acids) for each gene. The gene names were mapped to mouse genes. The feature f_COSMIC_ was defined as rank (number of mutations/length of gene). For genes with no COSMIC entry, the smallest value of f_COSMIC_ was used.

### 2.12. The Total Model (Linear)

The total model was set up as #citation = m + ∑_i_ c_i_ f_i_. Features were first scaled to have unit variance and zero mean. The intercept was discarded. The model was fitted in R using the limma package [12]. Because the features were normalized, we here report the raw coefficient values.

### 2.13. The Total Model (Nonlinear)

Several neural network models were fitted using the PyTorch library [13]. To avoid overfitting, we only considered networks with low numbers of layers. We picked one representative model, with 2 RelU layers (16 parameters); for example, having 3 RelU layers give similar output. Parameters were then searched using the ADAM optimizer (convergence shown in Appendix A). The relative importance of the features was estimated using a LIME [14]-like approach; for each feature and for each data point (gene), the neural network was asked to predict #citations if one standard deviation was added. The average difference in #citations was taken as the indicator of importance. The neural network explanation for T cells is shown in Appendix A. The RMSE was 0.55.

We also tested an approach based on gradient boosting (XGBoost) [15], resulting in an RMSE of 0.51. The model was tested in the same manner as the neural network (Appendix A).

The Jupyter notebooks containing the non-linear models are provided in the Github repository.

### 2.14. Creation of Online Data Visualizer

The online visualizer is provided at http://data.henlab.org/genepub created with the Python framework Dash (available online: https://plotly.com/dash/, version 1.10.0). Most of the underlying data is stored in SQLite3 files, which enable data to be read efficiently upon need. The files were generated using the R package RSQLite.

### 2.15. Drug Availability Analysis

The XML database from DrugBank 5.0 was downloaded and parsed in Java [16]. This program took all drugbank-id records, looked for a gene-symbol record and all gene-name records within target-records. These human gene symbols were translated into mouse gene symbols and compared. We tried taking both all targets and just the first target. Both yield similar citation correlations and drugs-per-gene trends, but including all the targets emphasized GABA-ergic genes higher. This is the approach used for Appendix A.

## 3. Results

### 3.1. The Scholastic Gene Coverage Follows Matthew Principle under Cellular Context

For simplicity and because mouse (*Mus musculus)* is the common model organism of choice for human diseases, we only focus on mouse and human genes. No doubt, simpler organisms such as *Caenorhabditis C. elegans* have been used as well, but for our own focus on immunology, mouse is the main go-to species. We include both protein-coding and non-coding genes whenever possible. Furthermore, we only consider 1:1 orthologs. An overview of the output of papers over time for genes is shown along with the first description of each gene (Figure 1b). The output increased exponentially, with a marked increase after the end of WW2 and gene discovery surging after the discovery of the DNA double helix in 1953 [17]. The rate of gene discovery increased continuously, despite such basic methods such as restriction enzyme-based cloning appearing only in 1973 [18].

Using this data, we wondered if the Matthew principle was an emerging property observed at a systemic level or was highly context dependent. For that, we first investigated to what extent the Matthew principle might dominate in biology if we consider each gene as its own scientific area. The distribution of the total number of papers for each gene is shown in Figure 1c. It follows a log-normal-like distribution, which is not compatible with the general Matthew principle. We then attempted to focus on a narrower set of papers, such as those within a biological subfield, and wondered whether this strategy would allow us to detect this. We focused our attention on different cell types and searched PubMed with suitable keywords. When restricting the context of study to a particular cell type and attempting a fit to the Pareto distribution, papers were closely (but not entirely) Pareto distributed (Figure 1f,g). This is compatible with the Matthew principle, and subsetting for all cell types yielded similar results (Appendix A). However, a random subsampling of papers down to the number for just T cells showed that this was a function of the number of papers covered and not a cell type effect (Figure 1e). We performed all analyses both on the total number of papers and for several cell types, but obtained similar results; for simplicity, we use total paper counts in the remainder of this paper.

### 3.2. A Total Model Integrates Sources of Reinforcement

We deliberated about the potential biological factors that could help us explain and predict gene popularity throughout time. To compare them, we decided to integrate different sources of reinforcement (referred to as “features”) and quantify their relative importance. Scores of each source were generated from several databases (Figure 1a) to be described and commented on throughout this paper. We used a linear model as well as nonlinear machine learning models (neural network and adaptive boosting) over the features to predict the number of citations (log10(1 + number of citations)) of each gene (henceforth called “#citations”). The nonlinear models made slightly better predictions (RMSE 0.51 and 0.55 vs. 0.78) but similar explanations (Appendix A). However, nonlinear models suffered from several problems, most importantly overfitting (due to more parameters, as exemplified in Appendix A), and the explanations could also have a complex local nature (partially described by, for example, SHAP [19] and LIME [20]). For the ease of interpretation, we here present the simpler linear model. The relative weight of the features is shown in Figure 1d (for T cells; other cell types in Supplementary File S1) and the confounder matrix (i.e., how strongly two features are correlated) in Figure 1e. The time from the first citation of a gene was the strongest feature (f_age_), contributing to an effect of self-reinforcement. It accounted for approximately 25% of the total reinforcement. The other features are now further described in roughly the order of their impact.

### 3.3. Paralogy and Gene Family Linkage Are Strong Drivers of Gene Popularity

We suspected that knowledge about a gene would reinforce research about similar genes. Genes are commonly named in groups such as *Gata1*, *Gata2*, and so forth. We denoted *Gata1* as the founding gene. One feature (f_founder_) was the #citations of the founding member. We further defined the family index feature (f_FI_), e.g., 2 for *Gata2*. The impact of these is best understood from Figure 2a. The founding member was almost always the most cited gene, which further explains why f_FI_ was inversely proportional with #citations. One concern is that say, *Gata2* might have been studied before *Gata3*, and thus obtain more citations due to mere age. However, even when regressing out the age, the same pattern emerged (Figure 2a, yellow).

The naming of genes, such as for *Gata1–5*, is not systematic; it may vary from the genes sharing a protein domain, or whole gene sequence homology, to sharing KO phenotype. To focus specifically on whole-gene homology, we used BLASTp to create a graph of gene-coded amino acid sequence similarity (see methods). We then defined the feature f_homology_ as the average #citations of neighboring genes (Figure 2b). This feature turned out to be roughly twice as predictive as f_founder_, with which it was moderately correlated (Figure 1e). The strength of the prediction was possibly due to f_homology_ being applicable to all the genes, while not all genes were organized in families following the considered name convention.

### 3.4. The Chromatin Structure Influences Citations in Several Manners

The advent of DNA-focused sequencing techniques has allowed researchers to further investigate the interaction between proteins and DNA, as well as chromosomal structure and loci positioning. Certain loci have been studied extensively due to how the chromatin influences gene expression, such as the T helper cell type 2 (Th2) locus, containing several important cytokines [21]. These cytokines are also co-expressed in Th2 cells (IL4, IL5, and IL13). We wondered if the chromatin structure influences the number of publications. A plot of chromosome 7 shows that this is clearly the case (Figure 2c). Coloring by gene family names, here simply based on genes sharing prefix (e.g., Olfr*), showed how families have been treated differently—the olfactory receptors being an extreme case.

To capture the chromatin influence, we defined f_chromatin_ by a neighbor graph, including the closest 10 genes along the genome. This feature was a strong predictor but with a complex interpretation. Together with the pure homology measures, these made up 45% of the prediction. Due to gene duplication mostly happening locally, chromatin structure captured homology, paralogy, gene families (by naming), TADs (topologically associating domains), and thus also co-expression. Thus, this feature, while hugely important, was non-trivial to interpret.

### 3.5. Gene Expression Drives Citations

Next, we examined the potential impact of tissue-specific gene expression on citation prediction. We used the Tabula Muris single-cell (sc)RNA-seq data, spanning 20 different mice tissues, and within each cell type computed the rank of average gene expression as feature f_exp_. This feature showed a clear correlation with #citations (Figure 2d). This correlation also held across cell types (Figure 2e), although we assumed that a gene was studied in the “wrong” cell type context (i.e., not the cell type with the highest expression) if less than 25 papers existed. Otherwise, genes were generally cited more in cells in which they were more expressed, although numeral counter examples existed. Some example genes are highlighted in Figure 2e. One example of anticorrelated genes is the Yamanaka factor *Oct4* (octamer-binding transcription factor 4, Figure 2f), in which the interest in stem cells was the driver for research despite it actually being highly expressed in professional antigen presenting cells. However, the biological role of both too much and too little expression of *Oct4* can cause differentiation [22], highlighting that high expression does not always imply higher biological importance.

### 3.6. Disease Association and Essentiality Make Up 20% of All Citations

While we did not address the impact of funding for simplicity reasons, we wondered about disease relevance being a considerable gene popularity reinforcer from a biological perspective. Historically, cancer has been (and still is) one of the most relevant malignancies for human societies, only behind cardiovascular disease [23,24]. For this reason, and thanks to the large body of literature and resources available, we decided to consider cancer in our final model of reinforcement. From the genome-wide COSMIC cancer mutation database, we defined f_COSMIC_ from the number of mapped mutations normalized by protein length. Similarly, we defined f_GWAS_ from the EBI GWAS catalog, which however include all forms of genome-wide associated studies (diseases and human traits). Overall, f_COSMIC_ was twice as strong a predictor as f_GWAS_ (Figure 1d).

Essential genes are of interest as therapeutic targets in cancer treatment. They also have clear phenotypes, making them easy to spot in screens, thus facilitating their study and reporting. Here we were able to define the feature f_essentality_ from CRISPR–Cas9 screens in 324 human cancer cell lines from 30 cancer types [11]. This feature was easily seen to be positively correlated with the number of citations (Figure 2g), of about the same impact as the f_GWAS_. The total of f_GWAS_ + f_COSMIC_ + f_essentiality_ made up about 20% of all citations.

### 3.7. Gene Regulatory Networks (Grns) Are Weak Reinforcers

One potential source of reinforcement may be studying genes if they are working together with other known genes. Examples include protein complexes, kinase-substrate pairs, and genes downstream of a common transcriptomic program. To investigate this, we used co-expression of genes in scRNA-seq data from Tabula Muris [8], as it provides an unbiased baseline for what constitutes a transcriptomic program. A GRN was calculated for each annotated cell type separately, using k-nearest neighbors (kNN) in the Euclidean space of normalized gene expression. The feature f_coexp_ was defined analogously to f_homology_, as the average #citations of neighboring genes. Clusters of highly expressed genes could be found in the UMAP projection (Figure 2h).

Because correlation in expression level primarily reflects transcriptional networks, we also investigated networks as defined by protein–protein interaction (PPI). For this purpose, we used the almost genome-wide HuRI dataset [10]. Using this neighbor graph without modifications, we defined f_PPI_. It had some degree of correlation with f_founder_ and f_homology_ (Figure 1g), which we suspected is due to heterodimerization with closely related proteins; but despite this, it added additional information to the total model (Figure 1d).

Overall, f_coexp_ + f_PPI_ only made up 5–10% of citations. It is possible that most genes were first discovered in isolation and only then added to regulatory networks.

### 3.8. Social Reinforcement Has Increased over Time

To address whether the sources of reinforcements have changed over time, we plotted several features for genes vs. the first mention of a gene (Figure 3a). Since the 1960s, genes discovered have been increasingly more essential or related to cancer or disease. This trend turned in 1990, after which all measures started decreasing. However, f_FI_ increased, suggesting that biologists aimed to describe the remaining genes independent of disease relevance [25]. We fitted our model to 1970–1990 and 1991–2010 separately (Figure 3b). Consistent with the above, f_age_ increased drastically, but mainly by offsetting gene family membership as a strong indicator. Essentiality and expression level in fact somewhat increased in importance, suggesting that while less relevant genes have now been covered, their coverage may be better tuned to their relative importance.

## 4. Discussion and Conclusions

Our final model is shown in Figure 3c. From our analysis, self-reinforcement (the Matthew effect) has a large impact on which genes we study. This is in line with similar findings from past studies [1,26,27,28,29]. Surprisingly, the effect increased post-1990. It cannot be attributed to new sequencing methods (pyrosequencing emerging in 2005 [30]), possibly rather to expression microarrays in 1995 [31], but more so to the Human Genome Project that started in 1990 [32], culminating in 2001 [33] (mouse in 2002 [34]). We had expected self-reinforcement to be stronger in the early days, given how few genes were known, but it is possible that the first genes were discovered due to their importance for the model organism used for study (e.g., insulin already in 1921). For simplicity, we limited ourselves to 1:1 human–mouse orthologs. As hinted by Stoeger et al., genes initially studied on certain model organisms (especially *Mus musculus* and *Rattus norvegicus*) have had an enormous impact on their citation popularity over their human homologues. As our study is primarily focused on human genes and we wished to retain as many genes as possible, we did not include additional species. However, the availability of model organisms has also impacted gene popularity [1].

Our model also highlights the impact of gene expression and co-expression features on gene popularity, for both coding and non-coding gene transcripts. This outcome particularly enforces the idea that high-throughput methods like (sc)RNA-Seq, DNA-Seq, protein biology focused methods, and CRISPR screens are central tools in the generation of unbiased datasets. This translates directly into blurring the historical perspective of traditional “one gene at the time” research (especially since 1990) and broadening the field´s scope towards a more integrative, systemic, and less biased understanding of the biological question studied by the researcher. Genes are now prioritized better according to their relevance. It is possible that this spills over into social bias, with some research into a handful of well-recognized genes being promoted instead of broadening the attention towards emerging secondary players (not exclusively restricted to families) that most likely complete the explanation for the biological event studied by the researcher. Our f_coexp_ + f_PPI_ features attempt to capture the interaction between the queried genes and their important secondary players at different levels. The relevance of these features, however, seem to have a low impact on gene popularity, potentially highlighting one of the main limitations of this study. GRNs are highly powerful tools for biological stochastic behavior analysis [35].

According to our model, features like gene homology (in terms of intraspecific amino acid sequence similarity between gene products), the presence of pre-existing gene family founders, and gene index (within the same family) are key players in spurring researchers towards exploring further gene families. An example of this is the family of olfactory receptor (*Olfr*) genes. This has a direct impact on the attention that some genes receive (especially from a family perspective).

Interestingly, our model shows disease relevance and essentiality features to be relevant for gene popularity, hinting at a cryptic transition of the genomics (and related) field from an essentially exploratory perspective towards a more goal oriented and context driven strategy (fueled partially by the advent of drug-target discovery [36]). This could be due to differences in funding, among other factors [1,37]. We did not include data to further investigate the impact of the funding system, which also might indirectly affect recruitment and researchers interests. However, other similar studies have shown interesting social cues that are responsible for part of the explanation of some genes’ popularity, including funding [1]. Altered researcher’s behavior may also affect citations in unclear ways; for example, newer generations of scientists tend to switch between topics more frequently [38]. Would they focus primarily on the commonly known landmark genes if they were to move to a new topic? The exponentially increasing pace of publications (Figure 1b) and the concept of “least publishable unit” is likely to also alter behavior in ways not analyzed here.

We have here included several features that we suspected were important; more features can be constructed. Some properties may be difficult to capture, and some genes are akin to black swans—their importance relies on unlikely events. For example, the COVID-19 target *Ace2* is likely to obtain disproportional coverage with our model and emphasize f_age_. However, even if our features were poorly affected by false positives/negatives, this would not affect their behavior over time. That said, if gene citation or annotation style has changed over time, this is something that would negatively affect our model. Thus, the trends in Figure 2 can be considered quantitative, even if other comparisons are better seen as qualitative. Overall, interpreting the results requires thinking carefully about the meaning of the features. The co-expression and expression features are influenced by the choice of tissues sampled. Cancer relevance may not be well represented through f_essentiality_, as it was calculated through a CRISPR KO screen rather than CRISPR activation, biasing it toward one type of cancer gene. There are several other caveats in interpreting essentiality as a proxy for cancer relevance [39]. However, this just begs for a harder question: Why else did it surge in importance in the 1990s? Many of the top drivers according to GWAS and COSMIC seem to have come earlier (Figure 3a). Was essentiality the best driver that could be found, after having run out of other strong disease candidate genes? In this regard, our analysis opens for more questions than we can answer at this time.

One other limitation in our study is that we have not investigated the impact of a changing cell type ontology. To avoid this, we have subjectively stuck with the most popular cell types. For example, the T cell type has been broken down into subtypes, and CD4 T helper cells eventually came to include not just Th1 and Th2, but also Th17 and the still somewhat ignored Th9 (as judged by citation counts). In future work it would be relevant to study, for example, how new cell types are “populated” with new genes from their founding type.

The top genes after 1990 are, in descending order, *Pten*, *Mthfr*, *Pparg*, *Mapk1*, and *Tlr2*-genes familiar to many biologists or clinicians as they frequently appear in textbooks (or as part of a mentioned protein complex or pathway). It is hard to imagine how we would have approached biology if we did not have at least some reference points. However, the number of drugs targeting (or known to target) a gene correlates highly with the citations (Appendix A, r = 0.4, Pearson correlation on log scale). Thus, as the scientific field of biology has matured, we likely need to look past our “comfort zone of familiar genes” and better integrate regulatory networks to find new drug targets. Unbiased methods such as CRISPR screens and single-cell analysis are likely to be of help. To further guide colleagues toward poorly explored areas, we provide http://data.henlab.org/genepub, showing properties of genes and indicating if they appear understudied. We hope this work enables reflective analysis and enables us to focus where it matters the most.

## Figures and Tables

**Figure 1 genes-12-00319-f001:**
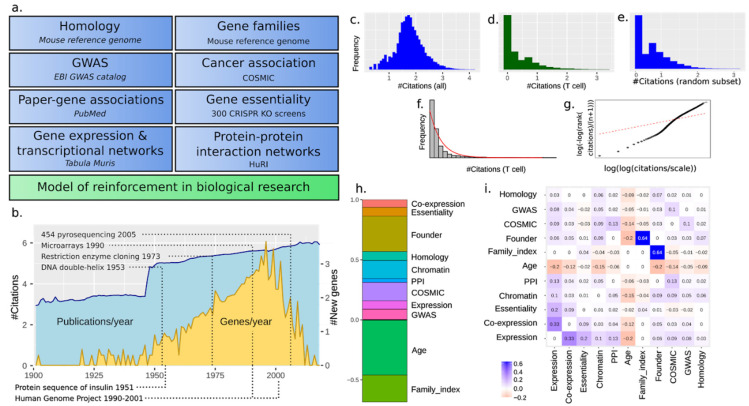
(**a**) Features defining the total model of reinforcement and the datasets used. (**b**) Citations and discovered genes over time, along with landmark events in genomic biology research. #Citations are log10(number of citations + 1). #New genes are log10 (number of genes + 1). Gene citation count distributions for all cell types (**c**) and for T lymphocytes (**d**). Overall gene publication frequencies follow a log-normal-like distribution, while when applying a cellular-type context, a Pareto-like trend appears. (**e**) Random subset of papers, similar to the number of papers for T cells in d. (**f**) A fit with an exponential distribution. This shows the super-exponential nature of citations. (**g**) Fit with the pareto distribution. This shows similarities but not a perfect fit either. We conclude that citations follow some intermediate distribution between these two cases. (**h**) Features and their relative weight contribution to the total fitted model. The order of features does not matter. Features do not sum to 1 but the input features are variance-normalized. Age is negative as we use the year of publication as a feature, which after normalization is negative age. (**i**) Spearman’s correlation coefficients between the included features for the total fitted model.

**Figure 2 genes-12-00319-f002:**
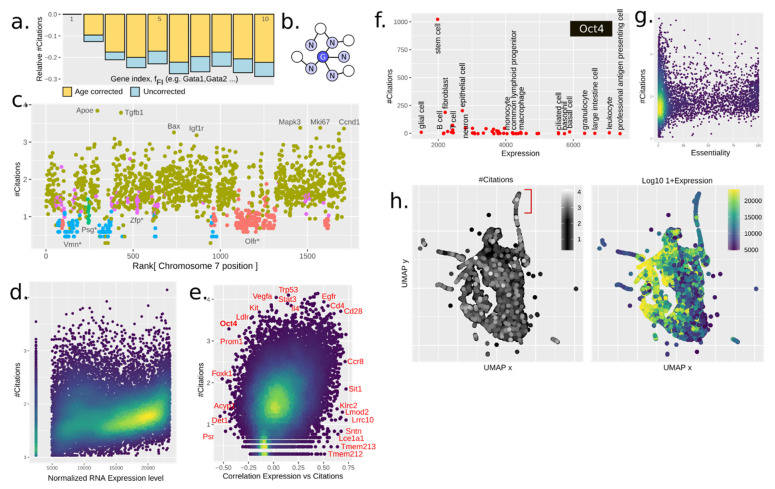
(**a**) Citation trends with relative #citations vs. gene family member’s index for the first 10 indices. Citations within gene families are normalized to index 1, which constitutes f_founder_. (**b**) Gene–gene graph representation. The average citations of neighboring genes in a gene–gene graph is used to define several features for each gene: f_coexp_, f_PPI_, and f_chromatin_. (**c**) f_chromatin_ defined as #citations vs. ranked chromosome position of gene, for chromosome 7, colored by some gene families named at the bottom*. Chromosomal position is a highly dimensional feature that captures several relevant biological parameters and strongly influences the #citations. Gene families tend to show similar patterns of citation. (**d**) #Citations vs. RNA expression level (primary tissue normalized) for T lymphocytes. Highly expressed genes generally tend to positively correlate with #citations. (**e**) #Citations vs. Pearson correlation values of RNA expression—#citations across cell types. The positive correlation trend observed in Figure 2d is consistent across cell types, thus reinforcing the idea of gene expression levels being a critical feature in gene popularity. (**f**) #Citations vs. cell type-specific expression level for the gene *Oct4*. Despite being highly expressed on professional antigen presenting cells, the cellular context in which *Oct4* has been extensively cited is stem cell research. This hints at the existence of underlying features not included within this study that might be paramount drivers of gene popularity. (**g**) f_essentiality_ defined as #Citations vs. cellular essentiality. Gene essentiality shows a positive correlation trend with gene popularity. This highlights that genes important for basic cell biology tend to produce a phenotype, which in turn facilitates gene reporting and enhances popularity. (**h**) UMAP projection of single-cell RNA-seq data showing the co-expression network (T cells, each point is a gene) colored by expression level and number of citations. A group of highly cited genes is pointed out in red.

**Figure 3 genes-12-00319-f003:**
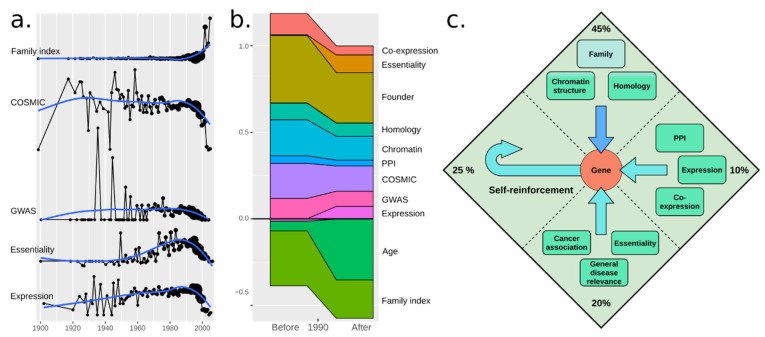
(**a**) Time series plot for several features where each point represents the year of first citation of a gene, from 1900 to 2010. Reinforcement sources for genes tend to vary over time, hinting at the existence of underlying social features for gene popularity (not included in this study), highly dependent on time. (**b**) Time series plot for all model-relevant feature weights for genes discovered between 1970–1990 and 1991–2010. Certain features like expression and essentially seem to be especially relevant as popularity determinators for genes discovered after 1990. (**c**) A summary of the proposed model of gene popularity reinforcers showing the total percentage of different sources of reinforcement.

## Data Availability

All the code is available on GitHub (https://github.com/henriksson-lab/genepub). The data can be browsed interactively at http://data.henlab.org/genepub.

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
