# Peer review of "Meta-Analysis of Gene Popularity: Less Than Half of Gene Citations Stem from Gene Regulatory Networks"

_genes, 2021, doi:10.3390/genes12020319_

Round 1

Reviewer 1 Report

The topic of the article is very interesting and fascinating: being aware of the possible biases that (scientific) knowledge may encounter is of fundamental importance for its general and harmonious development. Often one gets the impression that entire lines of research still suffer from historical choices (at the time justifiable) not adequately adapted and developed. Having critical and quantitative tools to be able to highlight and measure the possible deficiencies in certain sectors (genes and proteins considered less important or less "known" cellular and molecular contexts) is certainly very useful to help correct these inconsistencies, often not researched.

The paper presents certain results that are certainly useful, as well as interesting, but still presents points, in my opinion, to be developed and / or commented in more detail.

Specific comments:

  • Authors should stress more the fact of their “own focus on immunology”: it is said at line 91 but it is not so clear, and why?
  • At line 109, “We speculate that the overall log-normal distribution is an effect of the central limit theorem”: authors should be more precise and find a way to be more convincing about this conclusion.
  • Line 130, “For the ease of interpretation, we here present the simpler linear model.”: again, authors should be more precise and show some examples about the issues for nonlinear models related to these specific cases.
  • Looking for co-expression from GRN, authors have chosen co-expression of genes in scRNA-seq data from Tabula muris (line 229): would it be possible to have an historical perspective of the influence of co-expression (may be as from micro-array or RNA-Seq data) according to the historical gene study development? Like an interaction effect between GNR knowledge and time?
  • For disease association too, it would be interesting to explore the importance of gene-drug association mostly from a drug-repurposing perspective.

Reviewer 2 Report

Overall, this is a solid manuscript providing valuable insight into temporal trends in the research on genes. Further, it represents to my knowledge the first estimate of the extent to which the "social" contributes to research on genes - and thus sets an important hallmark for future work in the field. I hope that below suggests will further strengthen an already good manuscript.

When discussing the differences between log-normal and a more (but still not) Pareto-like distribution, the current interpretation emphasizes cell-types. This conclusion could possibly be strengthened by randomly sampling a corresponding number of publications from the shared log-normal distribution as the left-ward shift with a 0 mode would also be expected if considering fewer publications. Even if this shift was not due to cell-type, the manuscript would be interesting.

The text mentions that “genes are generally studied in the cells they are most expressed”, but does not provide support. The referenced Figure 2e appears to lend support for the weaker statement on genes generally being more cited if they are more expressed (but also does not provide a quantification of effect sizes across genes).

When the text shifts to cancer (section 2.6), I’m not sure if the subsequent analysis is general, or restricted to publications linked to cancer – either through text or genes. While either is good, a clarification would allow readers to better place the subsequent findigns.

The methods section leaves some ambiguity in how cell types were identified (e.g.: words of abstract, author-provided keywords, MeSH keywords; whether synonyms were considered)

Would the overall performance of the models that explain citations change over years? As present estimates seem to relate to the contribution of features to the models, it is unclear whether some features that appear less informative in recent years when compared to other features might actually have become more informative in absolute terms.

Other:

I’m unsure if I read Figure 1b correct and if #New genes should too be on a log-scale.

The current main text seems to bend the language around the Pareto-distribution to better fit theory – possibly to avoid conflicts with established views. Personally, I think that the Pareto is not necessary, and the in-between distribution in ED Figure 1b is an important finding that could go unnoticed with the current text.

The visualization of Figure 1d and similar panels, left me with several questions, that divert from the text: e.g.: consider area of individual colors or all blocks beneath; why does it stack to +1, but not -1?; is order important?;   with age being negatively does this mean an inverse relationship for year (so that oldest have lowest number) or with years passed?

Adaption of the Spearman correlation instead of Pearson in Figure 1e could possibly better account for nonlinearities.

The present analysis on single-cell gene expression compounds expression levels and cells with detectable expression. As several of the genes that the authors highlight in Figure 2e are inducible, a reader might be left wondering, if a distinction between housekeeping and situation-dependent genes would inform on citations.

Essentiality is currently introduced in the context of cancer. Though not uncommon, a word of caution toward the reader might be useful as the link between essentially and cancer has been flagged as one of the “Common pitfalls in preclinical cancer target validation”, Kaelin 2017, Nature Reviews Cancer.

The temporal analysis of 2.8 is exciting. Given the inverse U-shape in Figure 3a, it feels as if Figure 3b could be even stronger if replacing the two large bins by an annually progressing sliding window, or at least multiple different time windows.

For future works, I’d like to share with the authors my praise for https://github.com/titipata/pubmed_parser . This user-friendly Python library fulfils (and exceeds) many of the literature parsing tasks that the authors reimplemented.  

Round 2

Reviewer 1 Report

The authors have sufficiently addressed the points requested by me and included some, in addition to those proposed by the other reviewer, in the manuscript. Overall, I believe that some doubts have been clarified and some aspects better addressed.